# Individual-level surrogacy of MRI lesions for disease severity in RRMS: Methods to quantify predictive power and their application to longitudinal data from recent trials

Stefan Buchka[1]*, Joachim Havla[2], Begüm Irmak Ön[1,3], Raphael Rehms[1,4,5], Ulrich Mansmann[1]

1 Institute for Medical Information Processing, Biometry, and Epidemiology (IBE), Faculty of Medicine, LMU Munich, Munich, Germany, 2 Institute of Clinical Neuroimmunology, LMU University Hospital, LMU Munich, Munich, Germany, 3 Staburo GmbH, Munich, Germany, 4 Chair for AI in Healthcare and Medicine, Technical University of Munich (TUM) and TUM University Hospital, Munich, Germany, 5 Munich Center for Machine Learning (MCML), Munich, German

* sbuchka@ibe.med.uni-muenchen.de

## Abstract

### Background

Individual-level surrogacy (ILS) assesses how well a surrogate endpoint predicts treatment effect at the individual level. The paper discusses mutual information (MI) and the likelihood reduction factor (LRF) as ways to quantify ILS. It also reassesses ILS statements for people with relapsing-remitting multiple sclerosis (RRMS) by using T2 MRI lesions (T2L) as surrogates for disability and disease activity. ILS is often reported using inadequate concepts, e.g. subgroup analysis, correlation, odds ratios, sensitivity/specificity, or metrics based on Prentice's criteria.

### Methods

LRF assesses ILS quality by determining prediction quality through shared information (MI) between surrogate(s) and clinical endpoint(s). A simulation study validates LRF as a measure of ILS quality. Individual-level data from ten randomized controlled trials (n = 5673) provided longitudinal information on T2L, T2 MRI lesion volumes (T2V), future disability progression (EDSS), and relapses. LRFs for different scenarios were calculated. Results were compared to those obtained by methods commonly applied in RRMS literature.

### Results

Simulations confirmed the robustness of LRF as a reliable ILS quality measure. Two of ten trials showed weak ILS between T2V and EDSS (LRF = 0.21, CI95%: 0.16–0.26;

**Data availability statement:** The data is not publicly available but can be received via data requests from the Clinical Study Data Request (CSDR) website (https://www.clinicalstudy-datarequest.com) or Vivli website (https://vivli.org). This study used data from 10 clinical phase II and Phase III trials sponsored by Roche (3 trials), Sanofi (3 trials), and Novartis (4 trials). Trial details can be found via search tools of the repositories and the trial registration identifiers: NCT00050778 (CAMMS223), NCT00530348 (CAMMS323), NCT00548405 (CAMMS324), NCT00537082 (CFTY720D1201E1), NCT00289978 (CFTY720D2301), NCT00340834 (CFTY720D2302), NCT00355134 (CFTY720D2309), NCT01247324 (WA21092), NCT01412333 (WA21093), NCT00676715 (WA21493). Since some sponsors have in the meantime switched platforms (from CSDR to Vivli), not all studies are available via CSDR anymore. The specific CSDR data requests for this study can be found in the OSF repository (https://osf.io/ht4su/over-view). They were granted under the DUAs with CSDR proposal number 11223 (https://www.clinicalstudydatarequest.com/Posting.aspx-?ID=20600&GroupID=SUMMARIES). These data requests allow interested researchers to replicate the study findings in their entirety by directly obtaining the data from these third party sources and combining it with the published analysis scripts from the OSF repository, information given in the Methods section of the paper, and the Supporting Information files. In order to access the data researchers have to complete a data-use agreement with the related sponsors via the sharing platforms. The authors did not have any special access privileges; all datasets can be requested by any qualified researcher through standard procedures.

**Funding:** "Funding of Begüm Irmak Ön and Stefan Buchka (grant number: BMBF 01ZZ1804C) and Joachim Havla by DIFUTURE (grant number: BMBF 01ZZ1804B) by the Federal Ministry of Research, Technology and Space, Germany via the DIFUTURE (Data Integration for Future Medicine) consortium. The funders did not play a role in the study design, data collection and analysis, decision to publish, or preparation of the manuscript".

LRF = 0.28, CI95%: 0.23–0.34). Other LRFs were below 0.2. A method commonly used in the MS literature also showed no strong ILS.

## Conclusion

LRF is an important measure to quantify ILS and prediction quality. But it is rarely applied in RRMS research. LRF did not reveal robust surrogacy patterns when applied to data of ten clinical trials. Existing surrogacy claims should be reassessed since ILS assessments in the MS literature may have major limitations.

## Background

Treatment decisions for people with relapsing-remitting multiple sclerosis (PwRRMS) are based on clinical but also paraclinical disease activity [1–5]. MRI measures are often used to predict the disease course of RRMS and to design individual treatment strategies. MRI measures also indicate treatment failure during the future disease course [5]. Clinical decision-making can be based on individual-level surrogacy (ILS) concepts between a prognostic surrogate endpoint (SEP) and a clinical endpoint (CEP). Precise individual predictions of the future disease course (CEP) are essential. The paper explains that evidence on high-quality surrogacy cannot be reproduced within a set of randomized clinical trials.

Our paper discusses the likelihood reduction factor (LRF) as a measure to quantify ILS quality. LRF is related to mutual information (MI) [6] and is to our knowledge rarely applied in MS literature, but often discussed in the methodological literature on surrogacy. We explain why LRF is the method of choice and apply it to data from ten phase II/III clinical trials in PwRRMS.

If the SEP deteriorates in a PwRRMS, the treating physician may conclude that the future CEP is also deteriorating. Subsequently, he/she may suggests more effective treatment as a preventive measure. However, there is rarely reliable information on which physicians and PwRRMS can weigh up the pros and cons of the decision.

A less successful effort is underway to develop individual prediction models that are supposed to provide this information. Reviews in this field question the methodological state-of-the-art of models developed for PwRRMS [7,8]. Before outlining the rationale for choosing LRF as our preferred measure, we briefly review how ILS quality is addressed in the RRMS literature. One central weakness is the use of measures that capture associative effects instead of those that reflect individual predictive accuracy (see also section 4 in S1 Text).

Some papers [9–16] quantify ILS quality by demonstrating a significant difference in SEP between treatment response and non-response groups (as measured by the SEP outcome; see S1 Table for more details). Here, a significant p-value is considered proof of ILS. Unfortunately, it is not possible to convert the p-value into a quantitative risk statement. Significance provides qualitative evidence of a potential ILS effect and depends on the size of the population studied. A large population may even render a clinically irrelevant weak ILS effect significant. Additionally, authors

**Competing interests:** JH : reports a grant for OCT research from the Friedrich-Baur-Stiftung, Amgen/Horizon, Sanofi, Roche, and Merck, personal fees and nonfinancial support from Merck, Alexion, Novartis, Roche, Celgene, Biogen, Bayer, Neuraxpharm and Horizon/Amgen, nonfinancial support from the Sumaira-Foundation and Guthy-Jackson Charitable Foundation, all outside the submitted work. BIO: Since February 2025, BIO is employed in Staburo GmbH, a data science company with clients in the biopharma industry. The projects and clients that BIO is involved in are all outside of the scope of the submitted work.

**Abbreviations:** ARR: annualized relapse rate;AUC: area under the curve; CEP: clinical endpoint; CSDR: clinical study data request; EDSS: expanded disability status scale; FP: fractional polynomials; ILS: individual-level surrogacy; IQR: interquartile range;IT: information-theoretic approach; LR: log-likelihood ratio; LRF: likelihood reduction factor; MI: mutual information; MRI: magnetic resonance images; MS: multiple sclerosis; PTE: proportion of treatment effect explained; PwRRMS: people with relapsing-remitting multiple sclerosis; ROC: receiver operating characteristic; RRMS: relapsing-remitting multiple sclerosis; SEP: surrogate endpoint; T2L: number of new or newly enlarged T2 lesions; T2V: T2 lesion volume.

model the SEP-CEP relationship using logistic regression or Cox proportional hazards models. Typically, they do not report their models except for the odds ratio (OR) or hazard ratio (HR) attributed to the SEP [9–16]. Again, significant ORs or HRs are considered proof of ILS, even though the significance may depend solely on the size of the study sample. Clearly, large OR or HR values influence the predictive value of the SEP. The positive predictive value (PPV) for an individual is higher with larger OR values of the predictor. However, a large OR does not help to quantify individual risk precisely, as it reflects a relative ratio which transforms a baseline risk. Different individual baseline risks give different individually transformed risks even for the same OR. If we have a risk of 10% or 30% for a severe disease course in the group of a favorable SEP, the respective risks are 18% and 46% in the group of unfavorable SEP given an OR of 2. This missing link between OR or HR and risk statements is rarely reflected in MS literature.

One study provides correlation values between SEP and CEP [17]. However, a significant correlation is not sufficient to establish clinically relevant ILS quality. As we know from linear regression, the coefficient of determination ($R^2$) measures the percentage of variability explained. $R^2$ values close to 1 are essential for accurately predicting the dependent variable given the independent variable(s). The formula for $R^2$ is independent of the sample size used in the linear regression analysis. An informative report on the quality of ILS in linear regression settings should also include confidence intervals for $R^2$ measures.

Accordingly, we require a type of $R^2$ measure for more general settings between SEP and CEP. LRF is such a candidate and represents a generalization of $R^2$. However, the important point here is not whether the LRF is different from 0, but how high the LRF is. In order to make precise predictions, a threshold value must be determined to establish the quality and confidence intervals for the corresponding estimators. The threshold defines a minimal quality measure above which the metric is considered clinically useful. For example, an SEP with a corresponding LRF value above a pre-defined threshold may be indicative for treatment failure. Although correlation coefficients (r) and $R^2$ values are reported, the concept of applying pre-defined thresholds for clinical utility is not used in MS literature. To our knowledge, no studies to validate threshold values for r or $R^2$ has been conducted for PwRRMS. To define such a threshold for MS is out of the scope of this article.

When considering dichotomous SEP and CEP values (favorable/unfavorable), many papers report differences in SEP values between PwRRMS with favorable and unfavorable CEP responses. These differences are often expressed in terms of sensitivity (SE: percentage of people with an unfavorable SEP within people with an unfavorable CEP) and specificity (SP: percentage of people with a favorable SEP within people with a favorable CEP). However, sensitivity and specificity are not predictive concepts. Predictive statements are made using predictive values. The positive predictive value (PPV: percentage of people with an unfavorable CEP within people with an unfavorable SEP) and the negative predictive value (NPV: percentage of people with a favorable CEP within people with a favorable CEP) can be derived from sensitivity (SE) and specificity (SP), together with the prevalence

**Table 1. Trial information.**

| clinicaltrials.gov Registration number | Alternative Registration number | Analysis Groups[a] | Number of time points[b] | Duration (Enrollment – Completion) | Phase | N | Active Arm(s) | Control Arm |
|---|---|---|---|---|---|---|---|---|
| NCT00050778 [24] | CAMMS223 | CA1_1 CA1_2 | 4–5 | 5 years (2005/06/23–2010/01) | II | 333 | Alemtuzumab (12/24 mg) | Interferon β-1a (44 mcg) |
| NCT00530348 [25] | CAMMS323 | CA2_1 | 2–3 | 2 years (2007/09/13–2011/04) | III | 563 | Alemtuzumab (12 mg) | Interferon β-1a (44 mcg) |
| NCT00548405 [26] | CAMMS324 | CA3_1 CA3_2 | 2–3 | 2 years (2007/10/22–2011/09) | III | 798 | Alemtuzumab (12/24 mg) | Interferon β-1a (44 mcg) |
| NCT00537082 [27] | CFTY720D1201E1 | CF1_1 CF1_2 | 2–4 | 0.5 years (2007/09/27–2010/02) | II | 171 | Fingolimod (1.25 mg, 0.5 mg) | Placebo |
| NCT00289978 [28] | CFTY720D2301 | CF2_1 CF2_2 | 2–3 | 2 years (2006/02/09–2009/07) | III | 1272 | Fingolimod (1.25 mg, 0.5 mg) | Placebo |
| NCT00340834 [29] | CFTY720D2302 | CF3_1 CF3_2 | 1–2 | 1 year (2006/06/20–2011/07) | III | 1638 | Fingolimod (1.25 mg, 0.5 mg) | Interferon β-1a (30 mcg) |
| NCT00355134 [30] | CFTY720D2309 | CF4_1 CF4_2 | 2–3 | 2 years (2006/07/19–2001/06) | III | 1295 | Fingolimod (1.25 mg, 0.5 mg) | Placebo |
| NCT01247324 [31] | WA21092 | WA1_1 | 3 | 96 weeks (2010/11/23–2015/04/02 | III | 821 | Ocrelizumab (600 mg) | Interferon β-1a (44 mcg) |
| NCT01412333 [31] | WA21093 | WA2_1 | 3–4 | 96 weeks (2011/08/08–2015/05/12) | III | 835 | Ocrelizumab (600 mg) | Interferon β-1a (44 mcg) |
| NCT00676715 [32] | WA21493 | WA3_1 WA3_2 WA3_3 WA3_4 | 1–7 | 24 weeks (2008/05/09–2012/03/09) | II | 218 | Ocrelizumab (600/1000 mg) | Placebo/ Interferon β-1a (30 mcg) |

N; total number of participants included in trial; a: The groups analyzed refer to specific pairs of active and control treatment. For example, CA1_1 corresponds to the Alemtuzumab 12 mg (active) – Interferon β-1a (control) group and CA1_2 corresponds to the Alemtuzumab 24 mg (active) – Interferon β-1a (control) group; b: Longitudinal structures differ between endpoints. The available number of time points depends on the clinical and surrogate endpoint combination.

of unfavorable CEPs (see section 5 in S1 Text). For example, the title of Table 1 in MAGNIMS (2015) may be misinterpreted ("MRI criteria for predicting treatment response") as the data shown does not allow PPVs or NPVs to be calculated. In contrast, Table 1 in Rio et al. [15] enables such a calculation. Nevertheless, it is still not standard practice in ILS RRMS literature to report predictive values, although some papers report not only predictive values but also accuracy values.

Predictive values are conditional measures on sub-populations with favorable or unfavorable SEP outcomes. An ILS quality measure should assess ILS for the entire population, like the R² measure does. The accuracy percentage is a global measure for the entire population which is $SE \cdot P_{CEP} + SP \cdot (1 - P_{CEP})$, where $P_{CEP}$ is the prevalence of unfavorable CEP outcome. It is equal to $PPV \cdot P_{SEP} + NPV \cdot (1 - P_{SEP})$, where $P_{SEP}$ is the prevalence of unfavorable SEP outcome. However, accuracy cannot easily be generalised to settings involving non-binary endpoints and is rarely reported.

The literature also contains information on the quality of prediction models by reporting its area under the receiver operating curve (AUC under the ROC) [18]. The AUC is a discrimination measure. This information is not helpful when discussing precise individual risks during shared decision making. The quality of the prediction is also determined by calibration. Calibration of risk models assesses whether accurately predicted probabilities of risk events align with observed outcomes, ensuring the model's outputs are reliable for decision-making. Good calibration is essential to shared decision making. In the MS literature, information on model calibration is generally lacking. External validation is the state-of-the-art, but rarely provided for prediction models in the RRMS literature [7,8]. This means that the generalisability of the

evaluated prediction models to new persons remains unproven, and their clinical applicability cannot be assumed. We conclude that the availability of high-quality quantitative individual risk estimates is sparse.

A prominent assessment strategy for establishing ILS for PwRRMS uses the Prentice criteria [12] and the proportion of treatment effect explained (PTE), introduced by Freedman [19]: $PTE = \frac{\beta_1 - \beta_2}{\beta_1}$. Here, $\beta_1$ is the regression coefficient when CEP is regressed on the treatment variable, and $\beta_2$ is the regression coefficient for treatment when CEP is regressed on treatment and SEP. If SEP contains all (or none) of the information on treatment, then $\beta_2 = 0$ and PTE = 1 ($\beta_2$ = 1 and PTE = 0). PTE is a group-based measure derived from group-specific regression coefficients. However, it ignores the variability around the regression lines, which actually determines the quality of the CEP prediction given the SEP. Despite its frequent use [20,21], high PTE values do not guarantee accurate predictions at the individual level (see sections 1 and 4 in S1 Text).

Another qualitative approach to assessing the relevance of the SEP in a CEP prediction model is the likelihood ratio test (based on −2LLR, where LLR is the log-likelihood ratio) comparing models with and without the SEP. A significant test result is often considered evidence of ILS. However, even if the SEP has a small influence on CEP prediction, a large sample size may lead to significant test results. Again, a significant likelihood ratio test indicates potential ILS, but does not provide information on ILS quality.

Mutual information (MI) is a general concept that quantifies the amount of information shared between two random variables [6]. It is a powerful tool for detecting and measuring statistical dependence in fields such as machine learning, information theory, and data analysis. In likelihood models, MI is the mean log-likelihood difference between the model with and without the SEP: $MI = \frac{Deviance}{2N}$. In the ILS context, MI can be interpreted as the average change in individual likelihood of the model including the SEP compared to the model without the SEP. MI represents a form of likelihood alteration that is independent of sample size, and it quantifies the shared information between the SEP and the CEP at an individual level within the given population. In general, MI quantifies the increase in log-likelihood difference when independence between two outcomes is assumed.

Alonso and colleagues [6] introduced the LRF as a function of MI, as well as an instrument to understand the relationship between the SEP and the CEP.

$$LRF = 1 - e^{-2 \cdot MI(SEP, CEP)}$$
(1)

The mutual information (MI) between an SEP and a CEP is $MI(CEP, SEP) = H(CEP) - H(CEP|SEP)$. Here, $H(CEP)$ is the entropy of the CEP and $H(CEP|SEP)$ is the entropy of the CEP given the SEP. If the SEP predicts the CEP well, $H(CEP|SEP)$ is small compared to $H(CEP)$ and makes $MI(CEP, SEP)$ large, resulting in an LRF value closer to 1. If SEP and CEP are independent (SEP carries no information on CEP), the LRF equals zero since $H(CEP) = H(CEP|SEP)$. Alonso et al. demonstrated that the LRF has properties of a R² measure and equals R² in case of a linear model [6].

Section 3 of the S1 Text provides examples of how to calculate the MI or LRF for 2-by-2-tables (3a) or for a logistic regression model (3b). While the PPV only reflects those with an unfavorable SEP, the LRF applies to the entire population, regardless of the value of the SEP. An LRF close to 1 suggests that the examined prognostic factor(s) add relevant information to a predictive model. A high MI suggests that predictive models may exist, but whether such a model with high performance can be found depends on how the information is structured and exploited by the modeling approach. Exploring the LRF should be part of any predictive model development.

From a clinical perspective, the LRF expresses how much percentage of the uncertainty in an individual prediction is explained by considering the SEP. A higher reduction means that the prediction becomes more precise, for example when the expected range of a deterioration in EDSS values is narrower instead of wide. As mentioned above, a low variability in individual predictions is essential for informed clinical decision making.

We demonstrated the need for predictive models for PwRRMS. Many researchers have taken up this challenge, claiming to possess such models [7,22]. Often, however, the published models yield low to moderate performance metrics. One reason could be that the models cannot be any better due to the limited methodology used.

The paper is organised as follows: 1) We introduce the methodological concepts and present a simulation study to assess the LRF as a quality measure for ILS; 2) We analyse and interpret ILS in the data of ten MS trials; 3) We evaluate ILS as discussed in the literature; 4) We perform PTE analyses with our data and compare them to the results in the literature; 5) As a practical application we provide an assessment of the evidence of T2L as an individual-level SEP and treatment failure indicator based on literature cited in two MS guideline articles.

## Materials and methods

### Description of the trial populations

The Clinical Study Data Request (CSDR) repository [23] provided the individual data from three phase II and seven phase III randomized controlled trials (Table 1) on MRI measurements of T2 lesions, data on EDSS and relapses from adult PwRRMS (CSDR proposal number 11223). Four ILS settings (SEP: T2 lesion count or T2 lesion volume; CEP: EDSS or relapse count) were analysed in treatment vs control comparisons listed in S2 Table.

### Statistical analysis

Longitudinal CEP data was modelled by Gaussian or counting data regression models including or excluding longitudinal SEP as covariate (Equation (1)) and adjusting for treatment. The likelihood reduction factor (LRF) was calculated according to section 2 in S1 Text. The LRF values range between zero (no surrogacy) and one (deterministic dependency).

The SEPs studied are 1.) the log-transformed continuous T2 lesion volume (T2V) and 2.) the number of new or newly enlarged T2 lesions (T2L). The clinical endpoints (CEPs) are 1.) disability progression measured by the EDSS, which is treated as either a continuous or an ordinal outcome, and 2.) the annualized relapse rate (ARR) defined as the number of relapses per year.

In the count data regression models, we adjusted for time via an offset term (log(time)). We applied variance stabilizing transformation to normalize the count data [33,34].

As sensitivity analyses, we replicated ILS analyses according strategies applied in the MS literature and focused on three scenarios: 1.) *longitudinal predictive*: repeatedly measured SEP during the first year of the trial and repeatedly measured CEP during the second year; 2.) *aggregated associative:* SEP and CEP values were aggregated over the entire trial duration (for T2V change from baseline, for T2L total sum over a period); 3.) *aggregated predictive*: SEP (for T2V change from baseline, for T2L total sum over a period) at months 6 and 12 predicts the relapses (CEP) during the second year (S1 and S2 Figs). T2 lesion counts were truncated at eight.

Prentice criteria [35] were assessed and the PTEs were quantified (see section 1 in S1 Text) for relevant SEP/CEP combinations.

### Surrogate validation methods

Alonso and colleagues [6] proposed to estimate mutual information (MI) by the mean contribution of the log-likelihood ratio between two models. The first model includes the longitudinal SEP as a covariate, and the second model does not (Equation (2)).

$$mC\left[E\left(CEP_j\right)\right] = f\left(X_{C_j}^T, \beta\right) + \in_{C_j}$$

$$mC|S\left[E\left(CEP_j|SEP_j\right)\right] = g\left(X_{S_j}^T, \alpha\right) + \gamma SEP_j + \in_{S_j},$$

(2)

where:

- $CEP_j^T = \left(CEP_{j1}, CEP_{j2}, \ldots, CEP_{j1_p}\right)$ = Clinical endpoint vector

- $SEP_j^T = \left(SEP_{j1}, SEP_{j2}, \ldots, SEP_{j1_p}\right)$ = Surrogate endpoint vector

- $j = 1, 2, \ldots, n$ = number of subjects

- $X_{S_j}^T$, $X_{C_j}^T$, $\alpha$ and $\beta$ are the associated design matrices and trial treatment effects

- $\gamma$ are the trial specific effects of the SEP

- $mC$ and $mC|S$ link functions relate the linear predictors of the models to the mean value of the outcome.

- $f$ and $g$ are functions to describe the time trajectories

Following Equation (1), LRF $= 1 - e^{-2MI(SEP,\ CEP)} = 1 - e^{-\text{Deviance}/N}$.

Kent et al. and Alonso et al. [36,37] describe asymptotic confidence intervals for LRF. See section 2 in S1 Text for more details. The LRF was estimated for each pair of control and active study arm combinations (considering multi arm trials) as given in column *Analysis Groups* in Table 1.

## Data preparation

Observations collected during the open label extension of trials NCT01247324, NCT01412333, and NCT00676715 were excluded, as the removal of blinding in these phases could have influenced both participant-reported and investigator-assessed outcomes, including T2 lesion counts and EDSS measurements, potentially introducing bias. Depending on the SEP/CEP combination, pairs of longitudinal data were recorded by 5672 (T2V/EDSS), 5673 (T2V/relapses), 5672 (T2L/EDSS), or 4773 (T2L/relapses) PwRRMS. No imputation was performed and observations at the same measurement time point were excluded if either SEP or CEP were missing. EDSS values measured during and 90 days after a relapse were not used. PwRRMS with only one visit or without baseline visit were excluded. In SEP/CEP combinations using relapses as the CEP, the relapses occurring between two consecutive T2L or T2V measurements were summed for each PwRRMS and assigned to the later time point of the respective interval. S3 and S4 Figs present flow charts at observational and patient levels.

## Simulation study

SEPs and CEPs were simulated at two or four time points ($k = 2\ or\ k = 4$). In case of a continuous SEP, the data generating process was multivariate normal with mean zero, a standard deviation of one at each time point, and an auto-regressive correlation of $\phi = 0.8$. The SEP variance-covariance matrix results as $\Sigma_{SS} = D \cdot R_S \cdot D$, where $R_S$ is an auto-regressive auto-correlation matrix with $\phi = 0.8$, $D = 1_{k \times k} \cdot \in$, and $\in = 1$. A continuous CEP was generated by CEP(t) $= \alpha$ SEP(t) $+ \in$ (with $\alpha = 0.1, 0.5$, and $2.25$) plus an auto-correlated error term $\rho$ with $\phi = 0.8$ and standard deviation of $\in = 1$. This determines $\Sigma_{TT} = (\alpha^2 + \in^2) \Sigma_{SS}$ as well as $\Sigma_{ST} = \Sigma_{TS} = \alpha \cdot \Sigma_{SS}$ and results in $\Lambda = \left[\frac{\in^2}{\alpha^2 + \in^2}\right]^k$. We quantify how accurate the IT approach estimates the known $R_\Lambda^2 = 1 - \Lambda = LRF$ [38]. The simulation implements groups with 100, 300 and 600 patients. Barbiero and Ferrari describe how to simulate correlated Poisson distributed outcomes [39]. Huber's method [34] transformed count data for SEP and CEP to approximate normality to use Gaussian models for analysis. Detailed descriptions of how data is simulated for the Gauss-Poisson, Poisson-Gauss, and Poisson-Poisson case are given in S2 Text.

To examine the effect of a gradually emerging dependency $\alpha$ on $R_\Lambda^2$, we simulated SEP and CEP as uncorrelated during the initial measurement period and correlated later motivated by a scenario where early T2 lesion activity (SEP) can not predict disability progression (e.g., EDSS as CEP), but prediction may emerge over time when pathological processes accumulate. The correlation between SEP and CEP was set to zero for the first half of the measurement time

points. The auto-regressive correlation of $\varphi = 0.8$ of the SEP and CEP were simulated over multiple time points (ranging from $k = 2$ to $k = 50$) with varying $\in$ to assess the impact of parameters $\in$ and the number of time points $k$ on $R^2_\Lambda$ in. In cases with an odd number of time points, the first $\frac{k}{2} - 0.5$ time points were set to 0.

## Implementation

In the regression models (Equation (2)), potential non-linear trajectories over time as well as time by treatment interactions are modelled by fractional polynomials (FPs) [40]. Model selection follows the AIC step wise approach.

We used Gaussian models if CEP data was continuous. We also treated the EDSS as ordinal and grouped its values into ordinal categories (0, 0.5, 1, …, ≥ 4). An ordinal (multinomial) regression model was fitted as additional analysis. We fitted multinomial, Poisson, negative binomial, and zero inflated Poisson models if CEP was the number of relapses. The annual relapse rate was also analysed as ordinal outcome with three categories. Two or more relapses were summarized in one category. Mixed models were fitted with the R-package *glmmTMB* version 1.1.6, ordinal models by the R-package *ordinal* version 2022.11.16. S3 Table summarizes the model strategies of all analyses.

**Software and reporting guidelines.** All statistical analyses use R version 4.2.2. The list of packages is in S3 Text. The code is provided on https://osf.io (https://osf.io/ht4su/overview). The language was polished by the *ChatGPT* (OpenAI, 2024) version *GPT-4o-mini* [41] and *DeepL*. *ChatGPT* was also used to create S1 and S2 Figs, and R-scripts/ Figures assigned to sections 1–5 within S1 Text. We also provide the *TRIPOD* reporting guidelines [42] for the validation of prediction models (S4 Table).

## Ethics approval and consent to participate

The ethics committee of Ludwig-Maximilians University in Munich reviewed this project and granted a waiver (project number 19–838 KB). This decision was based on the written informed consent (as stated in the original publications [24–32]) from all patients for the use and sharing of data in all trials included in this study, as well as the anonymized nature of the collected data.

## Results

### Simulation studies

Fig 1 displays the results of the simulation study (rows: two or four time points; columns: analysis strategies). Columns one to six show LRFs for transformed count data, the last three columns display LRFs for original count data. For each setting, the red horizontal line indicates the true LRF values. The top of each panel shows the number of numerically unstable estimations out of 1000 simulation iterations. The results demonstrate that the LRF estimation performs reliably, with only two instances of numerical instability observed (column six, rows one and five). The estimates of the LRFs slightly underestimated the true LRF, particularly for a small number of simulated individuals ($n = 100$ or 300) and two time points. In scenarios with low correlations, the LRF can become negative indicating that the reduced model without the SEP has a higher log-likelihood than the model including it, suggesting the SEP lacks detectable predictive ability.

Results from the simulation study, conducted with $n = 1000$ iterations. Red horizontal lines in the graphs indicate the true LRF values. The numbers displayed at the top of each panel correspond to the instances of convergence issues observed during the simulation study. To derive the LRF, Gaussian, Poisson, or ordinal models were employed. It is important to note that the results of the Poisson family were analyzed in their untransformed form. The study considered four combinations of SEP and CEP: 1) Gaussian – Gaussian, 2) Gaussian – (transformed) Poisson, 3) transformed Poisson – Gaussian, and 4) transformed Poisson – (transformed) Poisson. The datasets included 100, 300, and 600 subjects

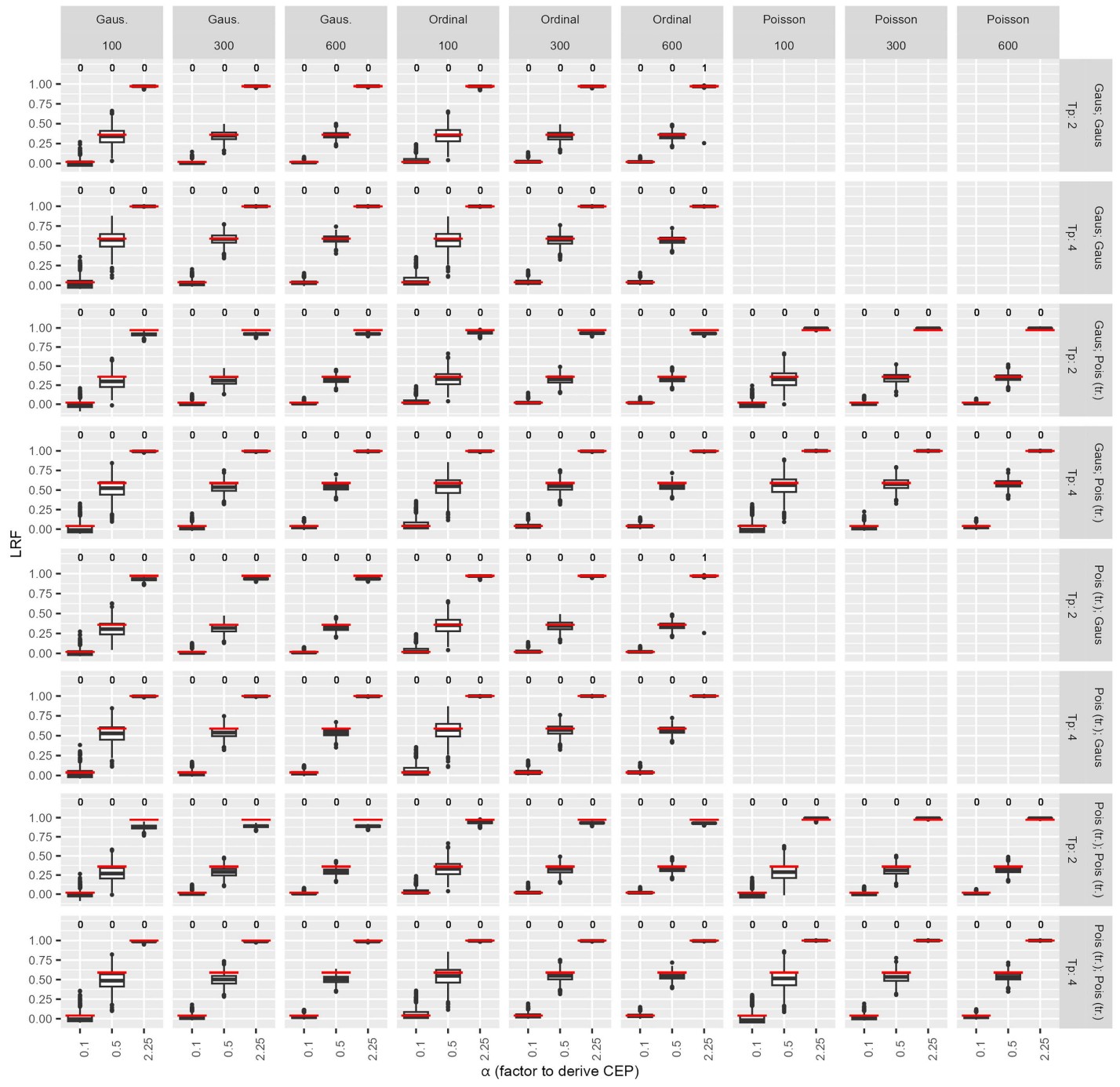

**Fig 1. Simulation (1000 iterations): LRF derived from models of different distributional families.**

(labeled at columns title), contributing to a comprehensive analysis of model performance. Abbreviations: LRF: Likelihood Reduction Factor; tr.: transformed; Pois: Poisson; Tp: time point.

S5 Fig presents additional results from the simulation scenario (SEP and CEP simulated as count data and analyzed using negative binomial and zero-inflated Poisson regression models). Instances of numerical instability were

observed. S6 and S7 Figs display the absolute differences between the true LRF values and those estimated from the simulated data. These differences range from −0.203 to 0.023 for original (S6 Fig) and from −0.104 to 0.023 for transformed count data (S7 Fig), indicating a small to moderate bias. When both SEP and CEP are normally distributed, the LRF shows minimal bias, though smaller sample sizes or transformed endpoints can underestimate its true value (S7 Fig).

The second simulation study uses the time-lagged model (S8 Fig). Results indicate that a higher number of measurement time points lead to higher $R^2_\Lambda$ (= LRF) values. $R^2_\Lambda$ is not adjusted to the number of longitudinal measurements.

### Surrogacy in the trial data

Seven of the ten trials were multi-arm trials, and 19 pairs of treatment/control arms were available for analysis (see Table 1). S5 Table provides median and interquartile ranges (IQR) aggregated over trial and time. During the trial, median changes in T2V ranged from −0.031 to −0.024, while EDSS showed a median change of 0. The mean relapse rate per time point ranged from 0.12 to 0.21, and the mean rate of T2L ranged from 1.45 to 1.46.

Fig 2 presents the LRFs derived from 19 pairings (Table 1 and S2 Table). The figure focuses on the four combinations of the SEPs and CEPs of interest, considering both transformed count data (columns one and two) and non-transformed count data (column three). For the T2V - EDSS outcome combination, LRFs for WA1_1 and WA2_1 exhibit values above 0.2, but all upper bounds of the LRF 95% CIs remain below 0.5, indicating a lack of strong evidence for ILS. Across all 68 scenarios, we observed two weak clinically non-relevant signals for ILS. The instabilities observed in Fig 2 (blue asterisks) indicate that the distribution of the number of relapses given T2V (P [Number relapses | T2L]) does not always perfectly follow a Poisson distribution. Since ordinal models are generally more prone to numerical instability, our results suggest that models with Gaussian-Gaussian endpoint combinations appear more reliable. Nevertheless, the observed numerical instabilities had minimal impact, as LRF values were highly consistent across model families with and without instability, showing comparable results between Gaussian, Poisson, and ordinal models (compare columns 1 to 3 in Fig 2).

LRFs were computed for 19 trial arm combinations (intervention vs. control). Gaussian, Poisson, or ordinal models were employed to derive the LRFs. Results from the Poisson model family were analyzed from non-transformed data. Four combinations of SEP and CEP (rows) were considered: 1) T2 Volume – EDSS, 2) transformed T2 lesion count – transformed relapses, 3) T2 Volume – transformed relapses, and 4) transformed T2 lesion count – EDSS. The calculated LRFs are represented by a point with 95% confidence interval bars. Blue asterisks indicate convergence issues encountered during the derivation of the LRFs from the corresponding models. Notably, the content in column 3 of Fig 2 and column 4 of S9 Fig, both utilizing the Poisson distribution family, is identical. Furthermore, the combination of T2V and EDSS outcomes remain consistent between Fig 2 and S9 Fig. Abbreviations: LRF: Likelihood Reduction Factor; tr: transformed; EDSS: expanded disease status scale.

The results for original count data (S9 Fig) align closely with those obtained for transformed count data. Convergence issues arose, particularly for the combination of T2V and relapse count, when using models with negative binomial distribution family.

### Sensitivity analyses

S10 Fig shows the results. The *aggregated associative*, *predictive longitudinal,* and *aggregated predictive* analyses show LRF values mostly around zero, but always below 0.2, indicating low ILS (columns one to four). Column 5 of S10 Fig shows the respective PTE values and their 95% bootstrap confidence intervals. The *aggregated associative* setting gives PTE ranges between −0.12 to 0.58 (mean: 0.21; median: 0.2) and the *aggregated predictive* setting shows PTE ranges between 0.01–0.36 (mean: 0.19, median: 0.17). The Prentice criteria are designed to assess surrogacy at a specific time point. An extension of the Prentice criteria to a longitudinal setting is not discussed in the literature.

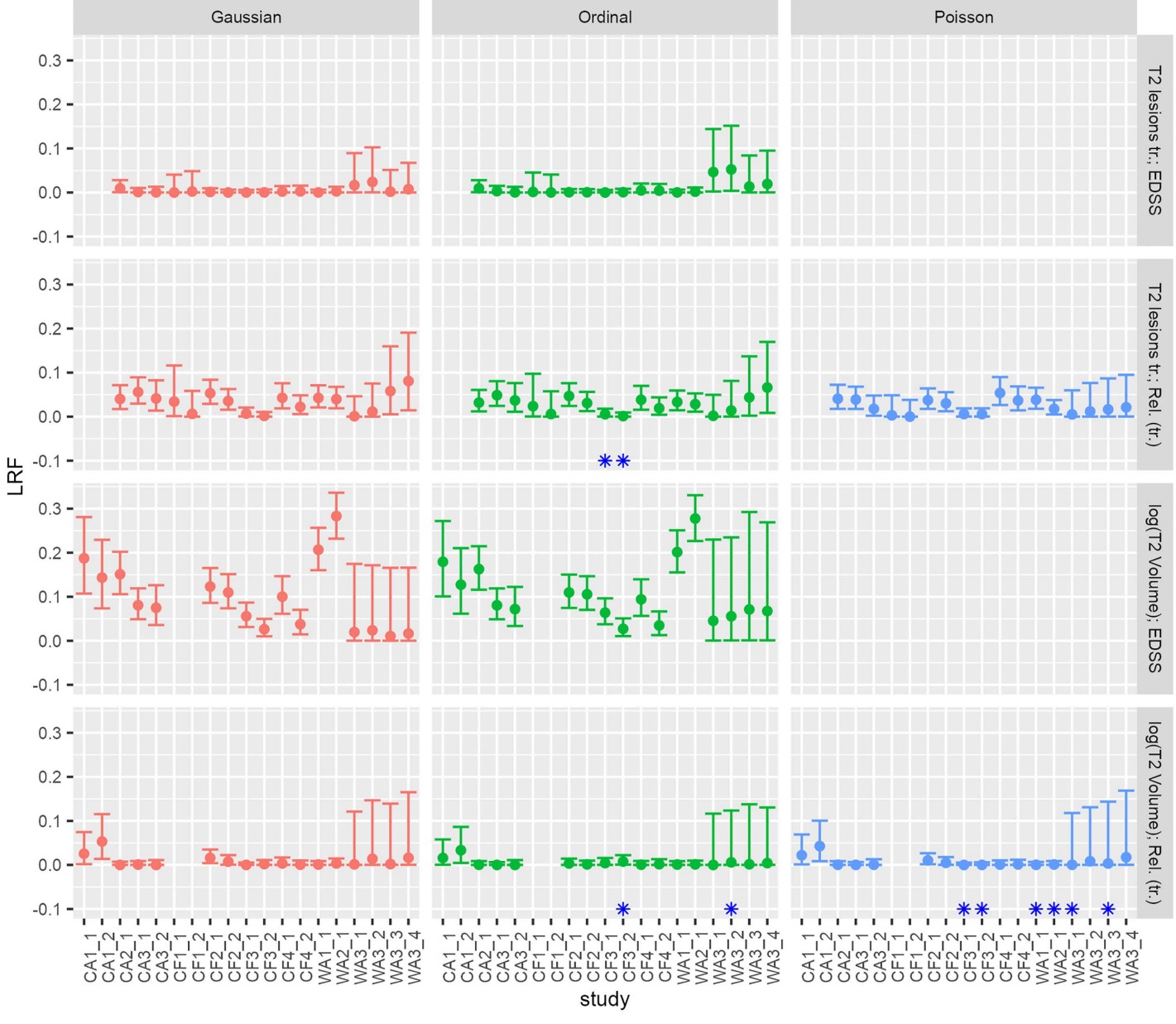

**Fig 2. LRF derived from different distributional families (transformed count data).**

All four Prentice criteria are fulfilled in three across the 13 *aggregated associative* SEP/CEP settings and in one across the nine *aggregated predictive* SEP/CEP settings (S10 Fig).

## Discussion

We focus on the likelihood reduction factor (LRF) as a quality measure for individual-level surrogacy (ILS). Currently, surrogacy between MRI-derived disease activity markers (SEPs) and patient-relevant clinical disease outcomes (CEPs)

is one of the central paradigms in the therapeutic decision-making process for people with remitting-relapsing multiple sclerosis (PwRRMS). With this work, we wanted to reproduce evidence for this established and recommended practice (see S1 Table).

We explored ILS in longitudinal independent individual-level data from 4773 to 5673 PwRRMS in three phase II and seven phase III trials. We could not find generalisable, robust scenarios with noteworthy ILS across trials (see Fig 2, S9, and S10 Figs). LRFs were mostly below 0.2 (Fig 2) indicating that in most examined scenarios less than 20% of the CEP variability is explained by the SEP. This means that the SEP provides relatively little information for predicting the CEP on the individual level, and such a level of explained variability may be insufficient to support informed treatment decisions. To provide an intuitive understanding of the LRF, Figs A and B in S1 Text may serve as illustrative examples. While data variability is low in Fig A and high in Fig B, this difference is reflected in the R² and LRF. Overall, the scenario shown in Fig A provides a more suitable basis for individual predictions than that in Fig B.

As shown in S7 Fig, this limited predictive power should also be interpreted in light of potential bias. While scenarios with transformed or non-normal endpoints tend to underestimate the true LRF (up to −0.104 for results corresponding to Fig 1), the bias is minimal when both SEP and CEP are approximately normal, and sample sizes are moderate to large ($N \geq 300$). Accordingly, the LRF values for T2V and EDSS (Fig 2, row 3) are not biased, while values in other scenarios may be underestimated. Even so, LRFs remain low. Overall, the results suggest that Gaussian-Gaussian endpoint combinations provide unbiased and less numerically unstable estimates compared to Poisson or ordinal models.

Another question is, if there is an appropriate cut-off point to differentiate between strong and weak ILS? In oncology, IQWIG (the German Institute for Quality and Efficiency in Health Care) has defined LRF values below 0.50 as being clinically irrelevant [43]. However, future research has to establish a clinically meaningful LRF threshold to differentiate between clinically relevant and irrelevant ILS for PwRRMS.

First, imagine a simple 2×2 table where 20% of the people have an unfavorable SEP. In this example, the PPV is 63% and the NPV is 90%, similar to cases reported by Rio et al. [15]. In this situation, the LRF is around 0.30, indicating a moderate reduction in uncertainty when predicting outcomes based on SEP. Second, consider how SEP is related to the CEP. Suppose CEP at time t depends on α times the SEP plus some random variability: $\boldsymbol{CEP}(\boldsymbol{t}) = \boldsymbol{\alpha} \cdot \boldsymbol{SEP}(\boldsymbol{t}) + \boldsymbol{\epsilon}(\boldsymbol{t})$ representing a regression model, where α is the effect of SEP on CEP and σ is the variability of $\in(\boldsymbol{t})$. If the variability is small compared to $\alpha$, the SEP is informative, and the prediction is more precise. If the variability is large, SEP explains little about the clinical endpoint and prediction quality is low (see sections 1b and 4 in S1 Text). For example, with four time points, the LRF can be calculated as $1 - \left[\frac{1}{1+\rho^2}\right]^4$, where ρ is the ratio of the SEP effect $\alpha$ to its variability. An LRF of 0.2 corresponds to a high variability ($\boldsymbol{\sigma} > 4 \cdot \boldsymbol{\alpha}$), while LRF = 0.5 corresponds to a smaller variability $\boldsymbol{\sigma} \approx 2.3 \cdot \boldsymbol{\alpha}$. Third, consider a logistic regression model predicting the probability of an unfavorable clinical endpoint based on SEP and additional patient information **X**. Examples in sections 3a and 3b in S1 Text show that ORs of predictors between 2 and 5 result in an LRF values of about 0.25, illustrating that even moderately strong predictors on the population-level only partially reduce uncertainty in individual predictions.

Results based on the whole population or subgroups are very different and must be considered differently. Our ILS assessment of ten independent clinical trials gave low LRF values. We showed that in the MS literature on individual prediction, mainly population-wide association measures were used. In most cases significant PTEs, ORs, and HRs (S1 Table) were reported instead of metrics suited for assessing the quality of predictions at the individual level, such as PPVs, NPVs, calibration, correlation, or the LRF. The unsuccessful search for population-wide prediction models may be explained by our weak ILS signals of the predictors used to forecast specific RRMS clinical outcomes. To avoid being misguided in their search for prediction models by too optimistic and inappropriate ILS statements, we recommend that predictive research in the field of MS focus on state-of-the-art ILS quality measures to assess settings on which future individual prediction models may be built. The use of PTE or the Prentice Criteria is misleading and can distort ILS assessments. See sections 1 and 4 in S1 Text or elsewhere [6,38].

The Prentice criteria and the PTE became more prominent. Two studies [20,21] assessed PTE in PwRRMS treated with interferon-beta-1 for the accumulated T2L values as SEP and the cumulative relapses as CEP (*aggregated associative*): PTE = 0.53 (0.28–1.01). Investigating first-year change in T2L as SEP and the cumulative relapses within the second year of treatment (*aggregated predictive* setting) resulted in PTE of 0.8 (0.34–1.86). Our data (data set W2_1, see Table 1) gave a PTE of 0.58 for the aggregated associative setting and PTEs below 0.8 for the *aggregated predictive* analysis (S10 Fig). By using comparable designs within the available trial data, we were unable to replicate these findings which have been seminal for ILS understanding in RRMS.

The analyses of prediction research that exist to date obviously influence guidelines and their application in everyday clinical practice. For example, the MAGNIMS guidelines recommend specific high-quality MRI T2L scans to monitor MS disease activity since T2L should have a high prognostic power for disease activity [1,2]. The CMSWG also recommends T2L as an indicator of treatment non-response, informing treatment decisions [1–4]. In a survey with US American neurologists, 97% (monitoring of RRMS within 12 months) and 67% (treatment switch to intravenous treatment, when ≥2 T2L occurred) of the neurologists follow these recommendations [44].

MAGNIMS and CMSWG recommendations claim a high predictive ability of T2L on relevant CEPs, based on various methodologies (see background). However, MAGNIMS and CMSWG also report a few predictive values which can be considered as ILS information. S1 Table provides a summary of our assessment.

From a clinical perspective, the utility of the LRF quantifying individual-level SEP quality can be illustrated in the following example: Consider a screening test with low sensitivity but high specificity. In this case, positive results are generally reliable (high specificity -> low false positives), but many true cases remain undetected, meaning the test alone is insufficient to guide decisions for all individuals. In a screening context, false positives can be clarified through subsequent testing before any intervention. In contrast, when a therapeutic decision is made directly based on this initial test, there is no opportunity to correct for misclassifications, which underscores the need for robust and reliable individual-level SEPs to guide treatment decisions. However, current evidence on T2L as a SEP detecting for treatment failure is frequently assessed using metrics that are not appropriate for prediction quality. Additionally, if a treated PwRRMS shows an unfavorable SEP and a treatment switch is recommended, it remains unclear whether the new therapy will be beneficial. There is uncertainty whether an unfavorable CEP will result from the unfavorable SEP and if the person responds to the new treatment. We found no guidelines discussing a benefit/risk trade-off for this decision-making process where information about the PPV of the SEP has to be combined with models that predict the best therapy to be switched to [45].

## Strengths and limitations

The study's strengths lie in its use of high-quality individual-level clinical trial data, a multi-trial analysis to identify ILS structures, and a state-of-the-art methodology applied to longitudinal observation structures within relevant RRMS populations.

Limitations include short to moderate observation periods of up to five years and the potential variability of T2L measurements and EDSS. Simple EDSS changes are more variable than confirmed disability progression, so results may be influenced by this. Our analysis focused on EDSS and relapse rate and did not consider composite or more modern endpoints (e.g., PIRA, cognitive measures, patient-reported outcomes), which may result in underestimating the true predictive value of MRI lesions and does not fully capture the pathophysiological complexity of RRMS. But follows the lines of clinical research which defines the present clinical work-up of PwRRMS.

T2 imaging standards across the trials were not established. However, internally, there were reading guidelines for MRI images. Accordingly, potential heterogeneity in T2 lesion detection and assessment between trials may be present. A limitation is the low incidence of events, such as T2L, in active trial arms. Other predictive MRI measures were not evaluated, nor whether existing methods sufficiently capture their predictive quality.

## Conclusions

We critically discussed methodologies commonly used in prediction research in the field of MS to claim ILS quality and explored ILS between MRI-derived markers and clinical outcomes within RRMS cohorts using independent individual-level data from ten MS trials. We examined ILS for T2L measurements and found weak signals in some trials, but no stable, generalisable ILS patterns between T2L as SEP for disease activity and disability progression in RRMS. To quantify ILS, we introduced the LRF as a reliable and generalized ILS quality measure for normally distributed longitudinal data. In some scenarios where endpoints followed a Poisson distribution, were ordinal, or had been transformed, the LRFs were underestimated by up to 0.104. However, this did not affect the overall interpretation of the LRF values derived from the ten clinical trials.

## Implications

Our findings underscore the complexity of predictive research in the field of multiple sclerosis. Based on our analysis, the clinical routine of making treatment decisions based solely on a new or enlarging T2 lesion during therapy should be questioned. However, we would like to clarify that our study does not question the clinical use of MRI or T2L monitoring but indicates that T2L provides limited predictive value for individual RRMS outcomes. But ILS for T2L measurements must be reassessed. We consider inadequate statistical methodology to be the main reason for ILS claims underlying relevant guidelines for treating and monitoring PwRRMS. We suggest that prediction research in MS should define appropriate standards to assess ILS. We argue that LRF is a good candidate for standardized reporting of ILS quality. A positive LRF assessment may make it worthwhile working on a prediction model which provides precise individual risk estimates for unfavorable CEPs. These models need development following a correct methodology and external, independent validation. To support the validity of guidelines, we recommend to report prospective risk estimates, validate models with calibration and discrimination metrics, define thresholds for predictive usefulness, and assess clinical utility via decision curve analysis in the future [46]. A consensus on such standards paralleling TRIPOD [47] would improve interpretability, timeliness, and relevance of MRI-based predictions and strengthen guideline recommendations. Research may focus on robust individual-level evaluation using appropriate metrics and methods to identify reliable SEPs for RRMS outcomes [23].

## Supporting information

**S1 Text. Diverse supportive information.**
(DOCX)

**S2 Text. Simulation study.**
(DOCX)

**S3 Text. List of R-packages.**
(DOCX)

**S1 Table. Summary of literature on T2 lesions as treatment response factor or surrogate endpoint in patients with relapsing-remitting multiple sclerosis.** Abbreviations: RCT, randomized clinical trial; PTE, proportion of treatment effect explained; OR, odds ratio; HR, hazard ratio; EDSS, expended disability status scale; ILS, individual-level surrogacy; TRiT, treatment response in treated; TRiuT, treatment response in untreated; T2L number of new/newly enlarged T2 lesion; MAGNIMS, magnetic resonance imaging in multiple sclerosis; CMSWG, the Canadian MS Working Group; CEL, contrast enhancing lesions; TLS, trial-level surrogacy; SEP, surrogate endpoint; DMT, disease modifying treatment.
(DOCX)

**S2 Table. Number of comparisons within trials for each SEP – CEP combination.** The table lists all combinations of surrogate endpoints (SEPs) and clinical endpoints (CEPs), each evaluated across all possible comparisons between active and control arms

within of each of the used trials. Since some trials include multiple active arms, more than one comparison per trial is possible. For surrogate validation, such comparisons are essential, as both a control and a treatment arm are required to assess whether the SEP is associated with the CEP, regardless of the treatment. Trial abbreviations correspond to those used in <u>Table 1</u> of the main paper. Trials marked in red provide only one of the two SEP types (either log-transformed total T2 lesion volume in cm³ or the count of new/enlarged T2 lesions). Abbreviations: EDSS, expanded disability status scale; SEP, surrogate endpoint; CEP, clinical endpoint. (DOCX)

**S3 Table. Model strategies.** The table displays the models utilized in both the main and the sensitivity analysis for both transformed and untransformed data. It is important to note that these model strategies were applied to both simulated and clinical trial data within the amin analysis. For details about the main analysis and the sensitivity analysis please refer to the methods part of the main article. Abbreviations: GLMM, generalized linear mixed model; EDSS, expanded disability status scale; SEP, surrogate endpoint; CEP, clinical endpoint; PTE proportion of treatment effect explained. (DOCX)

**S4 Table. TRIPOD Checklist for the validation of prediction models.**
(DOCX)

**S5 Table. Changes of endpoints over time.** For each SEP/CEP combination dataset, the table either presents the median (and interquartile range) of EDSS or T2 volume change from baseline, derived from aggregated trial data, or includes the median and interquartile range of the relapse or new/newly enlarged T2 lesion rate per trial visit. Abbreviation: IQR, inter quartile range; SEP, surrogate endpoint; CEP, clinical endpoint. (DOCX)

**S1 Fig. Different time association settings between SEP and CEP used in the sensitivity analysis.** A sensitivity analyses illustrating the individual-level surrogacy (ILS) were conducted by calculating the likelihood reduction factor (LRF) across different temporal association settings between SEP (new/enlarged T2 lesions - T2L - and T2 volume – T2V) and CEP (new relapses): longitudinal predictive (repeatedly measured SEP during the first year of the trial and repeatedly measured CEP during the second year) aggregated associative (summed-up SEP and CEP), and aggregated predictive (summed-up SEP at months 6 and 12, summed-up CEP within the second year). These analyses are restricted to data collected within the first two years of follow-up. Only trials with a duration of at least two years were included. (DOCX)

**S2 Fig. Different time associations between SEP and SEP.** Illustration of associative and predictive relationships between a surrogate endpoint (SEP) and a clinical endpoint (CEP) over time considering treatment. In the associative relationship (left), SEP and CEP are associated simultaneously, but values of former measurement time points influence following values (for example, the first value of the SEP influences the second one of the SEP and, when associated, also of the second value of the CEP). In contrast, the prognostic model (right) assumes SEP is measured before the CEP. The associative setting (left) corresponds to the main analysis described in the main article, whereas the predictive setting (right) represents the longitudinal predictive scenario considered in the sensitivity analysis. The trajectories of SEP and CEP are shown for both treatment and control groups over a 2-year period, highlighting differing temporal dynamics in both models. (DOCX)

**S3 Fig. Flow Chart on number of observations.**
(DOCX)

**S4 Fig. Flow Chart on number of patients.**
(DOCX)

**S5 Fig. Simulation (1000 iterations): Likelihood reduction factor derived from models with different families.**
Results from the simulation study (with n = 1000 iterations) utilizing the information-theoretic approach are presented. Red horizontal lines in the figures represent the true LRF values, and the numbers at the top of the panels indicate instances of convergence issues during the simulation study. To derive the LRF, Gaussian, Negative Binomial, Poisson, zero-inflated Poisson, and ordinal models were employed. Datasets with 100, 300, and 600 simulated subjects, each with two or four measurement time points, were generated. Four combinations of considered SEPs and CEPs were considered: 1) Gaussian – Gaussian, 2) Gaussian – Poisson, 3) Poisson – Gaussian, and 4) Poisson – Poisson. Abbreviations: LRF, Likelihood Reduction Factor; Gaus., Gaussian; NB negative Binomial; ZI. Poisson, Zero Inflated Poisson.
(DOCX)

**S6 Fig. Simulation (1000 iterations): Absolute difference between generated and true LRF using several model families.**
Results from the simulation study (with n = 1000 iterations) using the information-theoretic approach are presented. The displayed information represents the absolute difference between the generated and true LRF. To derive the LRF, Gaussian, Negative Binomial, Poisson, zero-inflated Poisson, and ordinal models were employed. Simulated datasets were created with 100, 300, and 600 subjects, each having two or four measurement time points. Four SEP – CEP combinations were considered, symbolized by different colors: 1) Gaussian – Gaussian, 2) Gaussian – Poisson, 3) Poisson – Gaussian, and 4) Poisson – Poisson. Abbreviations: LRF, Likelihood Reduction Factor; Gaus., Gaussian; NB negative Binomial; ZI. Poisson, Zero Inflated Poisson.
(DOCX)

**S7 Fig. Simulation (1000 iterations): Absolute difference between generated and true LRF (transformed outcomes).** Results from the simulation study (with n = 1000 iterations) using the information-theoretic approach are presented. The displayed information represents the absolute difference between the generated and true LRF. To derive the LRF, Gaussian, Poisson, or ordinal models were employed. Simulated datasets were generated with 100, 300, and 600 subjects, each having two or four measurement time points. It's important to note that results from the Poisson family were analyzed in their untransformed state. Four SEP – CEP combinations were considered: 1) Gaussian – Gaussian, 2) Gaussian – (transformed) Poisson, 3) transformed Poisson – Gaussian, and 4) transformed Poisson – (transformed) Poisson. Abbreviations: LRF, Likelihood Reduction Factor; tr., transformed; Pois, Poisson.
(DOCX)

**S8 Fig. $R^2_\Lambda$ when correlation between outcomes arises after the halve of measurement time points** . Abbreviations: CEP, Clinical Endpoint. $R^2_\Lambda$ is calculated in different correlation settings when the correlation between SEP and CEP emerges after half of the time points considered (correlation for uneven time points begins at the number of time points/2 - 0.5). Here, $\alpha$ represents the factor used to derive CEP, and $\in$ corresponds to the error in CEP, defined as $CEP = \alpha * SEP + \in$.
(DOCX)

**S9 Fig. Likelihood reduction factor derived from different distributional families.** Abbreviations: LRF, Likelihood Reduction Factor; EDSS, expanded disease status scale. LRFs were computed for each trial arm combination (intervention vs. control) using the information-theoretic approach by Alonso and colleagues. To derive the LRF, Gaussian, Negative Binomial, Poisson, zero-inflated Poisson, and ordinal models were utilized. Four SEP – CEP combinations were considered: 1) T2 Volume – EDSS, 2) transformed T2 lesion count – transformed relapses, 3) T2 Volume – transformed relapses, and 4) transformed T2 lesion count – EDSS. Blue or red asterisks indicate convergence problems of one or both regression models from which the LRFs were derived. Notably, the content in column 3 of Figure 2 and column 4 of Fig S9, both utilizing the Poisson distribution family, is identical. Furthermore, the combination of T2V and EDSS outcomes remains consistent between Figure 2 and Fig S9.
(DOCX)

**S10 Fig. Sensitivity analysis: Likelihood reduction factor derived from different distributional families.** Results of the sensitivity analysis are presented. LRF values derived by the information-theoretic (IT) approach and their 95% confidence intervals (CIs) are displayed in columns one to four, and the proportion treatment effect explained (PTE) values with 95% bootstrap CIs are shown in column five. Blue or red asterisks indicate model convergence issues within the IT methodology, while blue triangles indicate the fulfillment of the Prentice' criteria. Surrogate outcomes include new/newly enlarged lesions or log (T2 volume) change from baseline and the number of relapses as the clinical endpoint. The aggregated associative approach (rows one and four) evaluates the surrogate and clinical endpoint cross-sectionally after two years. The longitudinal predictive approach (rows two and five) considers surrogate measurements (T2 lesion count or T2 volume change from baseline) at months six and twelve, with the number of relapses within the second year as the clinical endpoint. The aggregated predictive approach (rows three and six) evaluates the surrogate (summed-up T2 lesion count or T2 volume change from baseline) within the first year and the number of relapses within the second year of the trial as the clinical endpoint. The IT approach was utilized to evaluate individual surrogacy. Only trials providing the necessary time structure are included. Abbreviations: LRF, likelihood reduction factor; PTE, proportion explained; ass., aggregated associative setting; long., longitudinal predictive setting; pred., aggregated predictive setting. T2 les., number of new/newly enlarged T2 lesions; T2 Vol., T2 volume; Rel., number of relapses.
(DOCX)

**S1 File. Responds to reviewes.**
(DOCX)

## Acknowledgments

This publication is based on research using data from "Sanofi", "Roche" and "Novartis" that has been made available through Clinical Study Data Request. We thank anonymous reviewers of the trials' sponsors for their helpful and constructive comments.

## Author contributions

**Conceptualization:** Stefan Buchka, Joachim Havla, Begüm Irmak Ön, Raphael Rehms, Ulrich Mansmann.

**Data curation:** Stefan Buchka.

**Formal analysis:** Stefan Buchka.

**Funding acquisition:** Ulrich Mansmann.

**Investigation:** Stefan Buchka, Ulrich Mansmann.

**Methodology:** Stefan Buchka, Ulrich Mansmann.

**Project administration:** Stefan Buchka, Ulrich Mansmann.

**Software:** Stefan Buchka.

**Supervision:** Joachim Havla, Ulrich Mansmann.

**Writing – original draft:** Stefan Buchka, Ulrich Mansmann.

**Writing – review & editing:** Stefan Buchka, Joachim Havla, Begüm Irmak Ön, Raphael Rehms, Ulrich Mansmann.

## Consent for publication

Before the publication, the sponsors of the trials reviewed the manuscript and consented with its publication.

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
