## [Decision Letter · Decision Letter 0]

16 Sep 2025

Dear Dr. Buchka,

Thank you for submitting your manuscript to PLOS ONE. After careful consideration, we feel that it has merit but does not fully meet PLOS ONE’s publication criteria as it currently stands. Therefore, we invite you to submit a revised version of the manuscript that addresses the points raised during the review process.

We look forward to receiving your revised manuscript.

Kind regards,

Simone Agostini, Ph.D.

Academic Editor

PLOS ONE

Journal Requirements:

2. Thank you for stating the following in your manuscript:

[Funding of Begüm Irmak Ön and Stefan Buchka via DIFUTURE (grant number: BMBF 01ZZ1804C) and Joachim Havla by DIFUTURE (grant number: BMBF 01ZZ1804B).]

[Funding of Begüm Irmak Ön and Stefan Buchka via DIFUTURE (grant number: BMBF 01ZZ1804C) and Joachim Havla by DIFUTURE (grant number: BMBF 01ZZ1804B). The funders did not play a role in the study design, data collection and analysis, decision to publish, or preparation of the manuscript]

[JH : reports a grant for OCT research from the Friedrich-Baur-Stiftung, Amgen/Horizon, Sanofi, Roche, and Merck, personal fees and nonfinancial support from Merck, Alexion, Novartis, Roche, Celgene, Biogen, Bayer, Neuraxpharm and Horizon/Amgen, nonfinancial support from the Sumaira-Foundation and Guthy-Jackson Charitable Foundation, all outside the submitted work.

BIO: Since February 2025, BIO is employed in Staburo GmbH, a data science company with clients in the biopharma industry. The projects and clients that BIO is involved in are all outside of the scope of the submitted work.]. 

We note that you may have received funding from these commercial sources: Sanofi, Roche, Merck, Alexion, Novartis, Celgene, Biogen, Bayer, Neuraxpharm

[JH : reports a grant for OCT research from the Friedrich-Baur-Stiftung, Amgen/Horizon, Sanofi, Roche, and Merck, personal fees and nonfinancial support from Merck, Alexion, Novartis, Roche, Celgene, Biogen, Bayer, Neuraxpharm and Horizon/Amgen, nonfinancial support from the Sumaira-Foundation and Guthy-Jackson Charitable Foundation, all outside the submitted work.

BIO: Since February 2025, BIO is employed in Staburo GmbH, a data science company with clients in the biopharma industry. The projects and clients that BIO is involved in are all outside of the scope of the submitted work.].   

We note that one or more of the authors are employed by a commercial company: Staburo GmbH.

5. We noted in your submission details that a portion of your manuscript may have been presented or published elsewhere. [It exists an pre-print of an older version of the manuscript:

Buchka S, Joachim H, Begüm IÖ, et al. Individual level surrogacy of MRI T2 lesion information for future disease severity: a methodological discussion and application to recent MS Phase II and III trials.

The reults presented in this manuscript will additionally published as a dissertation (monograph) of Stefan Buchka at the library of the Ludwig-Maximilians-University of Munich. The dissertation will not be publicly available before July 2026.]

Please clarify whether this conference proceeding or publication was peer-reviewed and formally published. If this work was previously peer-reviewed and published, in the cover letter please provide the reason that this work does not constitute dual publication and should be included in the current manuscript.

6. In the online submission form, you indicated that [Data are only available by an application to CSDR (Clinical study data request: https://www.clinicalstudydatarequest.com/)].

7. Your ethics statement should only appear in the Methods section of your manuscript. If your ethics statement is written in any section besides the Methods, please move it to the Methods section and delete it from any other section. Please ensure that your ethics statement is included in your manuscript, as the ethics statement entered into the online submission form will not be published alongside your manuscript.

8. Please include captions for your Supporting Information files at the end of your manuscript, and update any in-text citations to match accordingly. Please see our Supporting Information guidelines for more information: http://journals.plos.org/plosone/s/supporting-information .

Reviewers' comments:

Reviewer's Responses to Questions

**Comments to the Author**

1. Is the manuscript technically sound, and do the data support the conclusions?

Reviewer #1: Partly

Reviewer #2: Yes

2. Has the statistical analysis been performed appropriately and rigorously?

Reviewer #1: No

Reviewer #2: Yes

3. Have the authors made all data underlying the findings in their manuscript fully available?

Reviewer #1: Yes

Reviewer #2: Yes

4. Is the manuscript presented in an intelligible fashion and written in standard English?

Reviewer #1: Yes

Reviewer #2: Yes

Reviewer #1: This manuscript presents a methodologically rigorous but concerning challenge to established practices in multiple sclerosis research. While the statistical approach is sophisticated, there are several significant issues that warrant careful consideration:

Statistical Methodology Concerns

The authors' central argument rests on the superiority of the Likelihood Reduction Factor (LRF) over traditional surrogacy measures, but this premise has limitations. The LRF threshold of 0.5 borrowed from oncology may not be appropriate for MS, where the disease mechanisms and treatment effects differ substantially. The authors provide limited justification for why their chosen statistical measure should be considered the gold standard for clinical decision-making.

The simulation studies reveal concerning biases, particularly underestimation with smaller samples and numerical instability in multiple scenarios. Given that real-world clinical populations often have characteristics similar to the problematic simulation conditions, this raises questions about the reliability of the method in practice.

Clinical Relevance Issues

The most significant weakness is the disconnect between statistical surrogacy measures and clinical utility. The authors focus heavily on population-wide measures but provide insufficient discussion of how LRF values translate to individual patient care. A statistical measure showing weak surrogacy doesn't necessarily invalidate clinical utility if the surrogate still provides actionable information for treatment decisions.

The use of EDSS changes rather than confirmed disability progression is problematic, as acknowledged by the authors. EDSS changes are notoriously variable and may not capture the clinically meaningful disability outcomes that matter for long-term prognosis.

Study Design Limitations

The "short observation periods" mentioned by the authors are particularly concerning for a disease like MS where meaningful clinical outcomes may take years to manifest. The heterogeneity across trials (different imaging standards, observation periods, populations) introduces substantial noise that may obscure genuine surrogacy relationships.

The exclusion of patients with missing data rather than using appropriate imputation methods could introduce systematic bias, particularly if missingness relates to disease activity or outcomes.

Interpretation Problems

The authors' conclusion that "existing surrogacy claims should be reassessed" is overly strong given their methodological limitations. Their negative findings could reflect:

Inappropriate statistical measures for the clinical context

Insufficient observation periods for meaningful outcomes

Heterogeneity masking real relationships

The inherent complexity of MS pathophysiology

Missing Clinical Context

The manuscript lacks adequate discussion of how their findings should impact clinical practice. If MRI lesions have limited individual-level surrogacy by their measures, what alternative approaches do they recommend for treatment monitoring? The clinical community's widespread adoption of MRI monitoring reflects accumulated clinical experience that may not be captured by their statistical framework.

Reviewer #2: This is a well-structured and methodologically rigorous manuscript addressing an important topic in neuroimmunology and clinical trial methodology: the validity of MRI T2 lesions as individual-level surrogate markers for disease activity and disability in relapsing-remitting multiple sclerosis (RRMS). The authors introduce and validate the likelihood reduction factor (LRF) as a robust quantitative tool for assessing individual-level surrogacy (ILS), supported by comprehensive simulation studies and analyses of patient-level data from ten large clinical trials. The article has strong relevance for both clinicians and methodologists, and it makes an important contribution to the debate on how surrogate markers should be validated in neuroimmune diseases.

Major Problems

1. While the authors demonstrate that LRF is a reliable measure of ILS, the manuscript does not define a clinically meaningful threshold for interpreting LRF values in MS. Without a clear benchmark, it is difficult for readers to judge the clinical implications of “weak” vs “moderate” ILS signals. The discussion references oncology standards (IQWiG cutoff of 0.5), but a more disease-specific rationale would strengthen the paper.

2. The analysis relies primarily on EDSS and relapse rate. Both measures have known limitations (EDSS insensitivity to cognitive and subtle functional changes, relapse heterogeneity). The absence of composite outcomes or more modern endpoints (e.g., brain atrophy, patient-reported outcomes, cognitive measures) may underestimate the true predictive value of MRI lesions. This limitation should be more explicitly acknowledged.

3. The trials included did not follow standardized MRI protocols, which introduces variability in lesion detection. The paper notes this but does not quantify or adjust for it. Exploring how heterogeneity in MRI acquisition influences LRF estimates would add value.

Minor Problems

1. The rationale for excluding open-label extension data should be expanded.

2. Figures (e.g., simulation results and LRF estimates) are dense and may benefit from simplified summary panels or thresholds to guide interpretation.

3. The paper could highlight more explicitly how these findings should influence current clinical guideline recommendations (MAGNIMS, CMSWG).

4. “arouse” (p.17, line 373) → should be “arose.”

5. “proof that the LRF has properties” (p.8, line 176) → should be “proved” or “demonstrated.”

6. “PwRRMS” is introduced without definition in some places—ensure it is consistently defined as people with RRMS.

7. Some long sentences in the Background (pp.4–7) could be divided for clarity. For example:

8. References should be checked for consistent formatting (e.g., italics for journal names, spacing).

**Do you want your identity to be public for this peer review?** For information about this choice, including consent withdrawal, please see our Privacy Policy

Reviewer #1: No

Reviewer #2: No

---

## [Author Response · Author response to Decision Letter 1]

28 Oct 2025

Response to the Reviewers:

Dear Editors and Reviewers,

Thank you very much for the opportunity to revise and resubmit our manuscript. We sincerely appreciate the thoughtful and constructive comments provided by both the reviewers and the editorial team, which have been essential in improving the quality of our work.

In answer to the reviewers' feedback, we prepared a detailed, point-by-point response that outlines how each comment has been addressed. We also ensured that all journal guidelines and formatting requirements have been fully implemented in the revised manuscript.

Thank you again for your time and consideration. We look forward to your response and the opportunity to contribute to your journal.

Sincerely,

the authors

Comments of Reviewer 1 Answers/Changes in the manuscript

Statistical Methodology Concerns

The authors' central argument rests on the superiority of the Likelihood Reduction Factor (LRF) over traditional surrogacy measures, but this premise has limitations. The LRF threshold of 0.5 borrowed from oncology may not be appropriate for MS, where the disease mechanisms and treatment effects differ substantially. The authors provide limited justification for why their chosen statistical measure should be considered the gold standard for clinical decision-making. Statistical Methodology Concerns:

We appreciate the reviewer’s thoughtful comments on statistical methodology. In the revised manuscript, we expanded and clarified our rationale for using the Likelihood Reduction Factor (LRF) rather than frequently used surrogacy measures like OR, HR, sensitivity, specificity, or the ill-defined proportion of treatment effect explained (PTE). See text in the background for further clarification.

LRF is not an arbitrary chosen measure for the quality of prediction. It is the generalization of the well-founded R² measure (coefficient of determination) known to describe the extent of prediction error in linear regression. R² close to one is an excellent setting for prediction (nearly no prediction error) while R² of 0 is a fully random prediction setting. The LRF is already known as methodological concept around 2005 but never made it into the medical prediction literature. Our paper tries to provide this bridge and proposes to use LRF as a quality measure for surrogacy issues in medicine.

Introducing a threshold is a natural thing to do. We provide an example from the setting of added value regulation. The IQWIG defines R²=0.5 as the lower limit for useful surrogates in oncology. We support the statement of the reviewer that the MS community needs to find thresholds which serve their needs. Our future research will try to tackle the threshold problem by looking at concepts proposed by Wim van der Elst:

Van der Elst, Wim, et al. "The individual‐level surrogate threshold effect in a causal‐inference setting with normally distributed endpoints." Pharmaceutical Statistics 20.6 (2021): 1216-1231.

Why LRF is a useful metric:

We clarified in Line 114–117 that “The threshold defines a minimal quality measure above which the metric is considered clinically useful. For example, an SEP with a corresponding LRF value above a pre-defined threshold may be indicative for treatment failure” This shows that our use of LRF is intended to support a qualitative judgement on whether a factor is informative or not.

Our intention is to present the LRF as a relevant tool that contributes to the utility of SEPs in practice. By quantifying the reduction of uncertainty in individual predictions, the LRF provides clinicians with a clearer indication of whether a SEP meaningfully improves prognostic accuracy. This focuses on variability and clinical interpretability complements, rather than replaces existing clinical assessments in RRMS. We therefore added the following explanation in lines 204 - 209: “From a clinical perspective, the LRF expresses how much percentage of the uncertainty in an individual prediction is explained by considering the SEP. A higher reduction means that the prediction becomes more precise, for example when the expected range of a deterioration in EDSS values is narrower instead of wide. As mentioned above, a low variability in individual predictions is essential for informed clinical decision making.”

We further addressed the risk of misleading metrics by explicitly adding lines 96–100 “However, a large OR does not help to quantify individual risk precisely, as it reflects a relative ratio which transforms a baseline risk. Different individual baseline risks give different individually transformed risks even for the same OR. If we have a risk of 10% or 30% for a severe disease course in the group of a favorable SEP, the respective risks are 18% and 46% in the group of unfavorable SEP given an OR of 2.” In the discussion we added the following sentence (lines 492 - 496) “We showed that in the MS literature on individual prediction, mainly population-wide association measures were used. In most cases significant PTEs, ORs, and HRs (see Table S1) were reported instead of metrics suited for assessing the quality of predictions at the individual level, such as PPVs, NPVs, calibration, correlation, or the LRF.”. Furthermore, we reformulated a paragraph in the discussion (lines 472–489) to make the provided examples easier to interpret and assess.

Finally, we provide a schematic assessment of the metrics used in two relevant MS guideline articles on the clinical usability of existing treatment response tools. To emphasize this, we added a sentence in lines 218 - 220: “5) As a practical application we provide an assessment of the evidence of T2L as an individual-level SEP and treatment failure indicator based on literature cited in two MS guideline articles.”

On the use of thresholds:

In the revision we also emphasise (see lines 117–121) that, although correlation coefficients (r) and R² values are reported in the MS literature, this concept of applying pre-defined thresholds for clinical utility is also rarely used in MS literature. To our knowledge, no studies have validated appropriate thresholds for r or R² in MS. As stated in Line 120–121, “To define such a threshold for MS is out of the scope of this article.” We therefore do not propose a new threshold for R² or LRF values in MS but rather demonstrate that the LRF is a candidate measure because it represents a generalisation of R², which is (along with the r) used in MS literature. To help clinicians gain an intuitive understanding of this metric, we added the following sentence to the discussion (lines 449 - 457): “LRFs were mostly below 0.2 indicating that in most examined scenarios less than 20% of the CEP variability is explained by the SEP. This means that the SEP provides relatively little information for predicting the CEP on the individual level, and such a level of explained variability may be insufficient to support informed treatment decisions. To provide an intuitive understanding of the LRF, Figs A and B in S1 text may serve as illustrative examples. While data variability is low in Fig A and high in Fig B, this difference is reflected in the R² and LRF. Overall, the scenario shown in Fig A provides a more suitable basis for individual predictions than that in Fig B.”

On external validation

To further emphasize the importance of external validation of prediction algorithms, we added in the revised manuscript the following sentence (lines 154 - 157): “External validation is the state-of-the-art, but rarely provided for prediction models in the RRMS literature7,8.This means that the generalisability of the evaluated prediction models to new persons remains unproven, and their clinical applicability cannot be assumed” Our revision now makes clear that prediction models for RRMS are rarely externally validated, which limits their generalisability and clinical applicability. One objective of our study was to examine clinical trial data to determine whether the optimistic predictive claims frequently reported in the literature can be substantiated. See the following two guidelines on the development of validated prediction models:

Collins, Gary S., et al. "Evaluation of clinical prediction models (part 1): from development to external validation." Bmj 384 (2024).

Riley, Richard D., et al. "Evaluation of clinical prediction models (part 2): how to undertake an external validation study." Bmj 384 (2024).

By incorporating these clarifications, the revised manuscript (a) explains why the LRF is clinically informative, (b) clearly states that defining an MS-specific threshold is beyond the scope of our article, and (c) emphasises the importance of external validation.

The simulation studies reveal concerning biases, particularly underestimation with smaller samples and numerical instability in multiple scenarios. Given that real-world clinical populations often have characteristics similar to the problematic simulation conditions, this raises questions about the reliability of the method in practice. We thank the reviewer for pointing out the potential biases observed in our simulation studies. We therefore emphasized more clearly in the manuscript that, in our assessment, there is no systematic bias present in the simulation. We show bias in specific settings.

On the bias

Figure S7 corresponds to the results shown in Figure 2. When both SEP and CEP are normally distributed (Gaussian-Gaussian combinations; Figure S7 columns 1-3, rows 1-2), the bias is minimal, as the median of the simulated LRFs closely matches the true value. Variability is higher in scenarios with smaller sample sizes or moderate associations between SEP and CEP.

When either the SEP or CEP endpoint was transformed (e.g., for count variables such as T2L or relapse counts), bias increased, and the true LRF values were underestimated by approximately 0.1 (see the median LRF values in columns 1–3, rows 4–8). Similar patterns were observed for ordinal endpoints (columns 4–6). Using a Poisson model for non-transformed count endpoints resulted in underestimation only when both SEP and CEP were counts (columns 7–9, rows 7–8).

Applying these observations to Figure 2: for T2 lesion volume and EDSS, both endpoints are approximately Gaussian (column 1, row 3), corresponding to the scenario with the lowest bias in our simulations. Even in this case, LRFs were mostly below 0.15 and always below 0.3. Furthermore, most of the 10 included studies have sample sizes above 300, with six studies above 600 (range 333–1638; see Table 1), corresponding to medium and large sample size scenarios in the simulation. Even studies with N = 1638 show LRFs below 0.1 (see Figure 2, CF3_1 and CF3_2). LRF values remain low even in scenarios with bias and potential underestimation of up to 0.104 having minimal impact on the overall interpretation.

Overall, these results suggest that while small sample sizes or non-normal endpoints can lead to bias, the method remains informative, and in practical settings with moderate to large sample sizes and approximately normal endpoints, the LRF provides a reliable measure for assessing the predictive value of SEPs.

To clarify these points, we added the following text in lines 383-385 (Results), 458–464 (Discussion), and line 567 - 572 (Conclusion):

“When both SEP and CEP are normally distributed, the LRF shows minimal bias, though smaller sample sizes or transformed endpoints can underestimate its true value (Fig S7).

“As shown in Fig S7, this limited predictive power should also be interpreted in light of potential bias. While scenarios with transformed or non-normal endpoints tend to underestimate the true LRF (up to -0.104 for results corresponding to Fig 2), the bias is minimal when both SEP and CEP are approximately normal, and sample sizes are moderate to large (N ≥300). Accordingly, the LRF values for T2V and EDSS (Fig 2, row 3) are not biased, while values in other scenarios may be underestimated. Even so, LRFs remain low.”

“To quantify ILS, we introduced the LRF as a reliable and generalized ILS quality measure for normally distributed longitudinal data. In some scenarios where endpoints followed a Poisson distribution, were ordinal, or had been transformed, the LRFs were underestimated by up to 0.104. However, this did not affect the overall interpretation of the LRF values derived from the ten clinical trials.”

On numerical instability

The numerical issues shown in Figure S5 (NB and ZI Poisson) arise from the simulation design, where data were first generated as Gaussian and then transformed into Poisson-distributed variables. As expected, negative binomial (NB - columns 4-6) or zero inflated (ZI Pois. – columns 13-15) analysis methods do not fit to the simulated data.

The instabilities observed in Figure 2, column 3 (results using Poisson models to the data) suggest that the distribution of the number of relapses given T2V (P [Number relapses | T2L]) does not always follow a Poisson distribution. Ordinal models, in general, are not prone to numerical instability. Nevertheless, numerical instabilities have minimal impact, as the LRF values across the model families used to derive them are similar (compare columns 1 with 2 and 3 in Fig 2).

We added the following text to the results part of the revised manuscript (lines 402 - 409): “The instabilities observed in Fig 2 (blue asterisks) indicate that the distribution of the number of relapses given T2V (P [Number relapses | T2L]) does not always perfectly follow a Poisson distribution. Since ordinal models are generally more prone to numerical instability, our results suggest that models with Gaussian-Gaussian endpoint combinations appear more reliable. Nevertheless, the observed numerical instabilities had minimal impact, as LRF values were highly consistent across model families with and without instability, showing comparable results between Gaussian, Poisson, and ordinal models (compare columns 1 to 3 in Fig 2).”; to the discussion we added following sentence (lines 464 - 466): “Overall, the results suggest that Gaussian-Gaussian endpoint combinations provide unbiased and less numerically instable estimates compared to Poisson or ordinal models.”

In general, we employ well established methodologies for surrogate endpoint evaluation (introduced before 2005) and conducted simulation studies (Figure S7) to assess potential bias under different endpoint distributions and sample sizes, a step not previously done for other commonly used metrics (e.g. PTE).

Clinical Relevance Issues

The most significant weakness is the disconnect between statistical surrogacy measures and clinical utility. The authors focus heavily on population-wide measures but provide insufficient discussion of how LRF values translate to individual patient care. A statistical measure showing weak surrogacy doesn't necessarily invalidate clinical utility if the surrogate still provides actionable information for treatment decisions. We agree with the reviewer on the importance of providing statistical measures which are related to clinical reality. However, as discussed in the background section of our paper, the MS literature is full of such disconnects. No paper mentions the clinical meaning of a significant correlation between an SEP and a CEP and no paper explains the clinical meaning of a significant OR or HR. For us the clinical connection is represented if the prediction is precise. A precise prediction of an unfavorable clinical event is actionable. One should ensure the event does not happen. Furthermore, our study does not invalidate the clinical use of MRI or T2 lesions for disease monitoring but suggests that over periods of up to five years, T2 lesions provide limited predictive value for individual disease progression or treatment failure in RRMS. Population-level trends may remain informative, yet reliable individual predictions require evaluation using PPV, NPV, calibration, AUC, and internal and external validation. While we do not propose alternative monitoring tools, our work highlights key methodological considerations for identifying robust surrogate endpoints. To emphasize this, we added the following two sentences to the implications (lines 577 - 579 and 592 - 593): “However, we would like to clarify that our study does not question

---

## [Decision Letter · Decision Letter 1]

16 Nov 2025

Individual-level surrogacy of MRI lesions for disease severity in RRMS: methods to quantify predictive power and their application to longitudinal data from recent trials.

PONE-D-25-44981R1

Dear Dr. Buchka,

We’re pleased to inform you that your manuscript has been judged scientifically suitable for publication and will be formally accepted for publication once it meets all outstanding technical requirements.

Kind regards,

Simone Agostini, Ph.D.

Academic Editor

PLOS ONE

Additional Editor Comments (optional):

Reviewers' comments:

Reviewer's Responses to Questions

**Comments to the Author**

Reviewer #1: All comments have been addressed

Reviewer #2: All comments have been addressed

2. Is the manuscript technically sound, and do the data support the conclusions?

Reviewer #1: Yes

Reviewer #2: Yes

3. Has the statistical analysis been performed appropriately and rigorously?

Reviewer #1: Yes

Reviewer #2: Yes

4. Have the authors made all data underlying the findings in their manuscript fully available?

Reviewer #1: Yes

Reviewer #2: Yes

5. Is the manuscript presented in an intelligible fashion and written in standard English?

Reviewer #1: Yes

Reviewer #2: Yes

Reviewer #1: Although the responses to the comments are very wordy, the authors responded very well. Thank you very much.

Reviewer #2: No more comments. The author has fully addressed the reviewer's concerns. The publication of this article is of great significance to the field of MS.

**Do you want your identity to be public for this peer review?** For information about this choice, including consent withdrawal, please see our Privacy Policy

Reviewer #1: No

Reviewer #2: No

---

## [Editor Report · Acceptance letter]

PONE-D-25-44981R1

PLOS One

Dear Dr. Buchka,

I'm pleased to inform you that your manuscript has been deemed suitable for publication in PLOS One. Congratulations! Your manuscript is now being handed over to our production team.

Kind regards,

on behalf of

Dr. Simone Agostini

Academic Editor

PLOS One